# Impact of Lymphadenectomy on Survival of Patients with Serous Advanced Ovarian Cancer After Neoadjuvant Chemotherapy: A French National Multicenter Study (FRANCOGYN)

**DOI:** 10.3390/jcm9082427

**Published:** 2020-07-29

**Authors:** Virginie Bund, Lise Lecointre, Michel Velten, Lobna Ouldamer, Sofiane Bendifallah, Martin Koskas, Pierre-Adrien Bolze, Pierre Collinet, Geoffroy Canlorbe, Cyril Touboul, Cyrille Huchon, Charles Coutant, Emilie Faller, Thomas Boisramé, Justine Gantzer, Martin Demarchi, Jean-Jacques Baldauf, Marcos Ballester, Vincent Lavoué, Chérif Akladios

**Affiliations:** 1Department of Gynecologic Surgery, Hôpitaux Universitaires de Strasbourg, 67091 Strasbourg, France; lise.lecointre@chru-strasbourg.fr (L.L.); emilie.faller@chru-strasbourg.fr (E.F.); thomas.boisrame@chru-strasbourg.fr (T.B.); jean-jacques.baldauf@chru-strasbourg.fr (J.-J.B.); cherif.akladios@chru-strasbourg.fr (C.A.); 2Department of Public Health, Hôpitaux Universitaires de Strasbourg, 67091 Strasbourg, France; michel.velten@unistra.fr; 3I-Cube UMR 7357—Laboratoire des Sciences de L’ingénieur, de L’informatique et de L’imagerie, Université de Strasbourg, 67081 Strasbourg, France; 4Institut Hospitalo-Universitaire (IHU), Institute for Minimally Invasive Hybrid Image-Guided Surgery, Université de Strasbourg, 67081 Strasbourg, France; 5Department of Gynecology, Hôpital Universitaire de Tours, 37000 Tours, France; l.ouldamer@chu-tours.fr; 6Department of Gynaecology and Obstetrics, Hôpital Tenon, AP-HP, 75020 Paris, France; sofiane.bendifallah@aphp.fr; 7Department of Gynecology, Hôpital Bichat, AP-HP, 75018 Paris, France; martin.koskas@wanadoo.fr; 8Gynecological Surgery Service, CHU Lyon-Sud, Pierre-Bénite, 69000 Lyon, France; pierre-adrien.bolze@chu-lyon.fr; 9Department of Gynecological Surgery, Hôpital Jeanne De Flandre, CHRU, 59000 Lille, France; p-collinet@chru-lille.fr; 10Department of Gynecologic and Breast Surgery and Oncology, Hôpital la Pitié Salpétrière, AP-HP, 75013 Paris, France; geoffroy.canlorbe@aphp.fr; 11Department of Obstetrics and Gynaecology, Centre Hospitalier Intercommunal, 94000 Créteil, France; cyril.touboul@gmail.com; 12Department of Gynecology, Centre Hospitalier de Poissy, 78100 Poissy, France; cyrillehuchon@yahoo.fr; 13Department of Surgical Oncology, Georges-Francois Leclerc Cancer Center, 21000 Dijon, France; charles.coutant@tnn.aphp.fr; 14Department of Medical Oncology, Hôpitaux Universitaires de Strasbourg, 67091 Strasbourg, France; justine.gantzer@chru-strasbourg.fr; 15Medical Oncology Department, Centre Paul Strauss, 67000 Strasbourg, France; MDemarchi@strasbourg.unicancer.fr; 16Department of Gynecologic and Breast Surgery, Groupe Hospitalier Diaconesses Croix, Saint-Simon, 75020 Paris, France; marcos.ballester@aphp.fr; 17Department of Gynecologic Surgery, Hôpital Universitaire de Rennes, 35000 Rennes, France; vincent.lavoue@chu-rennes.fr

**Keywords:** ovarian carcinoma, neoadjuvant chemotherapy, systematic lymphadenectomy

## Abstract

Background: The population of interest to this study comprised individuals with advanced-stage ovarian carcinoma who were exposed to neoadjuvant chemotherapy (NAC) followed by interval debulking surgery (IDS). Those who had not received systematic lymphadenectomy (SL; Group 1) were compared to those who had received SL (Group 2). Outcome measures included progression-free survival (PFS), overall survival (OS), and surgical complications. Methods: This was a retrospective, multicenter cohort study in nine referral centers of France between January 2000 and June 2017. OS analysis using the multivariate Cox regression model was performed. PFS and surgery-related morbidity were analyzed. Results: Of the 255 patients included, 100 were in Group 1 and 155 in Group 2. Patient majority was, on average, younger and less comorbid, with predominant R0 surgery in Group 2. Dindo–Clavien score was similar between the two groups (*p* = 0.15). Median OS was 26.8 months in Group 2 and 27.6 months in Group 1. SL was not statistically significant on OS (*p* = 0.7). Median PFS was 18.3 months in Group 2 and 16.6 months in Group 1. SL had positive impact on PFS (*p* = 0.005). Conclusions: patients who had received SL (Group 2) had significantly higher PFS regardless of node-positivity status when compared to those who had not received SL (Group 1).

## 1. Introduction

Ovarian cancer is the eighth leading cause of cancer in women. In France, it was estimated that 5193 new cases of ovarian cancer occurred during 2018, and the number of deaths was estimated at 3479 [1]. Approximately 75% of new cases are diagnosed at International Federation of Gynecology and Obstetrics (FIGO) Stage III or IV [2]. Five-year age-standardized net survival reached only 44% for cases diagnosed in the period of 2005–2010 [3].

The standard treatment for advanced ovarian cancers is currently based on complete tumor-cytoreduction surgery and chemotherapy based on platinum and taxanes [4]. The objective of this surgery was to obtain no residual tumor (R0), which means no macroscopically visible tumor remaining [5]. R0 is recognized as a major prognostic factor for increased overall survival (OS) and progression-free survival (PFS) [6,7]. If upfront debulking is unattainable, interval debulking surgery (IDS) after neoadjuvant chemotherapy (NAC) was accepted as an alternative in women with advanced-stage ovarian cancer [8,9]. Lymphatic spread was reported to be a common feature and important prognostic factor in both early and advanced ovarian cancer [10].

Regardless of the International Federation of Gynecology and Obstetrics (FIGO) stage of the patients, 44% to 53% of metastatic lymph nodes were found by systematic lymphadenectomy [10,11,12]. It is not yet clearly defined on systematic-lymphadenectomy (SL)-improved survival [13].

Some retrospective studies [14,15,16,17,18] suggested overall survival benefit, even with complete surgery (tumor residue of <10 mm) from systematic pelvic and paraaortic lymphadenectomy. In contrast, two prospective randomized trials [10,19] did not show improvement in overall survival, though one found increased progression-free survival [19].

Panici et al. [19] evaluated only the extent of lymphadenectomy, and included both patients with macroscopically complete resection and those with residual tumors of up to 1 cm in diameter after surgery; the LION trial [10] did not concern patients after neoadjuvant chemotherapy.

However, actual practice is heterogeneous, as many teams reported the use of lymphadenectomy at first surgery to realize staging after neoadjuvant chemotherapy for the removal of lymph nodes considered to be a reservoir of tumor cells that are difficult to completely eliminate by chemotherapy [20]. French recommendations published in 2019 [21] called for pelvic and paraaortic lymphadenectomy in advanced cancers in the case of the clinical or radiological suspicion of metastatic lymph nodes.

Given this conflicting evidence and the absence of a consensus on systematic lymphadenectomy, we conducted a retrospective and multicentric study to compare two groups of patients with primary serous ovarian carcinoma, benefiting NAC and followed by IDS: no lymphadenectomy (Group 1), and lymphadenectomy (pelvic and paraaortic; Group 2). Overall survival and PFS, complications related to surgery and lymph-node involvement were assessed.

## 2. Materials and Methods

We conducted a retrospective, multicenter cohort study in nine referral centers of France constituting the FRANCOGYN study group: Tenon hospitals, Jean Verdier, Créteil, Poissy and La Pitié Salpêtrière in Paris and the Parisian region, as well as the University Hospitals of Lille, Rennes, Tours, and Strasbourg. All women gave their consent to participate in the study. The research protocol was approved by the Ethics Committee for Research in Obstetrics and Gynecology (CEROG 2016-GYN 1003).

Analysis was performed of the medical records of all patients with histologically confirmed advanced epithelial ovarian cancer of Stages III or IV according to the FIGO classification with serous subtype, diagnosed between January 2000 and June 2017.

Among these patients, we included those who, on initial assessment, were considered unsuitable for complete resection (R0) due to the extent of their disease, and who benefited from interval debulking surgery (IDS) after platinum-based neoadjuvant chemotherapy.

The number of NAC cycles was variable. In fact, there are currently no standardized protocol and objective criteria for deciding the date of surgery on their basis. This was left to the discretion of the medical team, but also according to the availability of the technical platform of the center for IDS programming. An initial surgical evaluation, whenever realized, had to only be for diagnostic purposes (oophorectomy and/or peritoneal and omental biopsies); patients undergoing other procedures, and those who benefited from other types of treatment with chemotherapy, such as bevacizumab before or after surgery, were excluded. Nonepithelial ovarian tumors, other histological types than serous, and unoperated patients were also excluded.

The study population was divided into two groups: patients who underwent no lymphadenectomy (Group 1) and patients who underwent lymphadenectomy (Group 2), as presented in a flowchart (Figure 1). The following clinical and demographic variables were collected: age at diagnosis, body-mass index (BMI), personal or family history of gynecological cancers, personal comorbidities assessed by the Charlson comorbidity index [22], the American Society of Anesthesiologists (ASA) score [23], the presence or absence of identified genetic mutations and preoperative CA 125 level. Tumor characteristics were also detailed, namely, FIGO stage [22] and tumor grade. Lastly, therapeutic data were noted: number of cycles of NAC, and the used therapeutic. All operated patients had a satisfactory response to NAC on the scanographic study, and two subgroups were created separating partial and complete responses. We collected data on the cytoreduction procedure (laparoscopy, laparoscopy followed by conversion to laparotomy, and laparotomy). Surgery included at least hysterectomy, bilateral salpingoophorectomy, infragastric omentectomy and the removal of any other intraperitoneal metastasis in all centers and during the study period. The residual tumor was assessed at the end of IDS, and defined by complete resection (R0) when all visible tumors were removed (no macroscopic RD) at the end of the intervention, R1 when it was ≤2.5 mm, and R2 when it was more than 2.5 mm, but less than 2.5 cm. Lymphadenectomies and the assessment of the residual tumor were left to the discretion of each surgeon. Surgical staging was defined according to the FIGO staging system. The initial extent of the disease at the start of each surgical procedure was quantified using the peritoneal-cancer index (PCI) described by Sugarbaker [24]. Operative time, and the number of transfused packed red blood cells and plasma were reported, as well as the occurrence of per- and postoperative complications, evaluated by Dindo–Clavien classification [25].

Overall survival (OS) was defined as the time from the date of initial diagnosis to the date of death (of any cause). Patients with an unknown date of death at the end of the study were censored at the last date of contact. Progression-free survival (PFS) was defined from the date of initial diagnosis to the date of first tumor recurrence necessitating either medical or surgical intervention. Recurrence diagnosis was based on the increase of tumor marker CA-125. The site of recurrence was defined as pelvic, intraperitoneal, retroperitoneal, or distant metastatic.

Descriptive analysis of all included patients was performed. Patient characteristics, tumor characteristics, and operative findings by lymphadenectomy or not were compared using Wilcoxon’s test for quantitative variables and chi^2^ or Fisher’s exact tests for qualitative variables. Survival curves were constructed using the Kaplan–Meier method with the logrank test applied to detect differences between groups. Multivariate analyses for OS and PFS were performed using Cox regression models. Hazard ratios (HRs) and 95% confidence intervals (95% CIs) were estimated. All covariates with a *p*-value of less than 0.20 in univariate analysis were included in multivariate analysis. The proportional-hazard assumption was graphically verified. All statistical tests were two-sided, and the significance level was set at 0.05. Statistical analyses were carried out using SAS Studio (SAS Institute, Cary, NC, USA).

## 3. Results

During the study period, a total of 501 patients were assessed for eligibility (Figure 1). Of the 500 patients with advanced ovarian carcinoma with neoadjuvant chemotherapy alone, 245 patients were excluded: 227 because they had histological subtypes other than the serous subtype, and 18 because there were 2343 missing data on residual tumors (*n* = 8) and on whether lymphadenectomy was performed (*n* = 10). A total of 255 patients with all inclusion criteria were included in analysis and divided into two groups, 155 patients in the lymphadenectomy group (60.8%) and 100 patients in the non-lymphadenectomy group (39.2%).

Patients were balanced between the groups (Table 1) without any statistically significant difference except for age, ASA score, and Charlson index. In fact, patients who did not benefit from lymphadenectomy were statistically older (*p* < 0.001) and had more comorbidities in Charlson index (*p* = 0.003) and ASA score (*p* = 0.03).

In the lymphadenectomy group (Table 2), the proportion of patients with Stage III disease was significantly higher (81.9% vs. 78%; *p* < 0.0008) even if overall FIGO stage did not differ between the two groups (*p* = 0.4). There was no statistically significant difference between preoperative CA125 (*p* = 0.36). There was statistically significant difference between the two groups concerning the number of NAC cycles (*p* = 0.02) with a majority of patients benefiting from less than three cycles (16.7% versus 5.1%). Subgroup analysis of the NAC response showed a majority of the complete response in Group 2 (27.8% vs. 52.8%; *p* = 0.007).

Table 3 relates to surgical characteristics during IDS. A median of 28 resected lymph nodes were reported in Group 2, with 15 paraaortic and 13 pelvic nodes. Pathological diagnosis revealed microscopic lymph-node metastases in 11%, including 10% and 13% in pelvic and paraaortic metastasis nodes, respectively. The addition of lymphadenectomy to debulking surgery had a significant effect on the median duration of surgery (383 vs. 242 min, *p* < 0.0001). There were significant differences in the surgical approach for debulking surgery (*p* = 0.01).

However, when subgroup analysis was carried out, Group 2 mainly benefited from laparoscopy (33.3% vs. 16.3%; *p* < 0.0001), while Group 1 had more laparotomies (75.6% vs. 59.6%; *p* < 0.06). There was no statistical difference between the two groups concerning intraoperative laparoconversion (*p* = 1). There was no difference concerning patients receiving transfusions (median 16 vs. 9; *p* = 0.7), the number of transfused red blood cell units (median 2.2 vs. 2.9; *p* = 0.45), or fresh frozen plasma (median 1 vs. 0; *p* = 0.79).

The lymphadenectomy group also had higher on intra (20.9% vs. 6.5%; *p* = 0.003) and postoperative complications (26.7 vs. 12.8%; *p* = 0.001), but the Dindo–Clavien score was not statistically significant in either group (*p* = 0.15).

When calculated, Sugarbaker score, corresponding to the dissemination of carcinosis at the beginning of debulking surgery, was similar in both groups (median 25 versus 20 l; *p* = 0.44).

The macroscopic peritoneal residual tumor showed statistically significant difference between groups (*p* =< 0.0001). In Group 2, R0 rate was constant with a strong majority (88.4 vs. 50%; *p* < 0.0001), and a lower rate of R1 (5.8 vs. 23%; *p* = 0.02) and R2 (5.8 vs. 27%; *p* = 0.004).

With regard to postoperative systemic treatment, 85.6% of patients in Group 1 and 90.6% in Group 2 were treated with platinum and taxanes (*p* = 0.27), with a similar number of cycles between the two groups (median 6.5 vs. 6.9; *p* = 0.2).

Median OS was 26.8 months, with 26.8 months (95% CI, 21.6 to 36.4) in Group 2 and 27.6 months (95% CI, 20.7 to 36), in Group 1 (Table 4). The logrank test showed no significant difference (logrank = 1.5 (0.6–1.4), *p* = 0.73). In univariate Cox regression analysis, lymphadenectomy was not significantly associated with overall survival, with HR = 0.9 (0.6–1.4) *p* = 0.7.

Similarly, median PFS was 17.2 months, with 18.3 months (95% CI, 16.3 to 20.1), in Group 2 and 16.6 months (95% CI, 14.9 to 18.7) (Table 5) in Group 1, without any difference between groups (logrank 1.1 (0.8–1.5, *p* = 0.48)).

Kaplan–Meier curves are shown in Figure 2 and Figure 3 for OS and PFS, respectively. In multivariate analysis, CA125 level at diagnosis was a predictor for worse OS (HR = 1.9 (1.1–3.1), *p* = 0.01; Table 4). On the other hand, factors influencing PFS were lymphadenectomy regardless of the number of removed lymph nodes (*p* = 0.005) and surgery debulking, with subgroup analysis showing that laparoconversion had an adverse impact on PFS (HR = 154.8 (5.4–4411), *p* = 0.004; Table 5).

Concerning recurrence sites (Table 6), there were no pelvic recurrences in either group, and no statistically significant difference in intraperitoneal retroperitoneal and distant metastatic recurrences (*p* = 0.2; 0.67 and 0.76, respectively).

## 4. Discussion

In our study, patients with initially inoperable advanced serous-subtype ovarian cancer who had IDS after NAC had no significant difference in OS, but had improvement in PFS at five years between patients who underwent or not systematic pelvic and paraaortic lymphadenectomy whatever the number of removed lymph nodes (*p* = 0.005), and whatever their status (metastatic or not). This is the only study evaluating the impact of SL in IDS in patients with advanced serous ovarian adenocarcinoma after NAC. Other studies evaluated the impact of SL in patients with ovarian adenocarcinoma of all histological subtypes after NAC [26,27,28,29,30,31]. Our results are consistent with those published [26,27,29,30,31] that did not demonstrate improvement in OS for SL after NAC.

Eoh et al. demonstrated improvement in overall survival [28]. The major difference between our study and that of Eoh et al. [28] was that they compared two groups: lymph-node sampling (LNS) (<20 resected node) and lymph-node dissection (LND) (min 20 resected nodes), and not SL versus no lymphadenectomy. They showed improvement in OS with a median survival of 28 months for LNS vs. 37 months for LND, with HR = 0.29 (0.15–0.57) *p* = < 0.001. The same was true with PFS, with HR = 0.637 (0.429–0.946), *p* = 0.025 [28]. Concerning PFS, SL had positive impact. The majority of studies evaluating the impact of lymphadenectomy after NAC did not show a similar result [30,31], with the exception of two studies [27,28]. Song et al. [27] compared three groups: lymphadenectomy on suspicious nodes, SL, and nonlymphadenectomy. Median PFS was 28, 30.5, and 22 months, respectively, and nonlymphadenectomy was an independent factor affecting PFS (HR = 1.729, 95% CI 1.213 to 2.464, *p* = 0.002), with no difference between the other groups [27]. Previous studies showed that SL at the time of primary cytoreduction improved the progression-free-survival rate without any impact on overall survival [19]. Chan et al. published a retrospective study on 13,918 patients with Stage III–IV ovarian cancer from the Surveillance Epidemiology and End Results (SEER) database, and showed a benefit in terms of survival for systematic lymph-node removal. They demonstrated that an increase in the number of collected pelvic and lomboaortic lymph nodes was significantly associated with an increase in disease-free specific survival at five years, regardless of age, grade, number of metastatic nodes, and stage of disease. For a number of collected nodes (1, 2–5, 6–10, 11–20 and greater than 20), overall survival was 36.9%, 45%, 47.8%, 48.7% and 51%, respectively (*p* = 0.023). These results were also found in the study by Chang et al. [32].

Other studies showed that systematic lymphadenectomy at the time of primary cytoreduction was not associated with improved OS or PFS [10,33].

Currently, there are only two randomized studies available with high level of evidence [10,19], and the meta-analysis of three randomized studies [14]. Harter et al. [10] concerned patients with FIGO IIIB–IV, and it is the most recent. Median overall survival was 65.6 versus 69.2 months in lymphadenectomy vs. nonlymphadenectomy, respectively, with, in Cox multivariate, HR = 1.06 (0.83–1.34), *p* = 0.65 [10]. Disease-free survival was 25.5 and 25.2 months in lymphadenectomy vs. nonlymphadenectomy, respectively, with HR = 1.11 (0.92–1.34), *p* = 0.29 [10]. Du Bois, et al., in exploratory analysis of three prospective randomized trials investigating chemotherapy regimens in advanced ovarian cancer with no macroscopic residual tumor, showed significant impact of lymphadenectomy on overall survival [8]. Similarly, there are four meta-analyses in the literature with divergent results. Gao et al. [34] identified 3488 subjects, and the five-year overall survival rate in the LND group was higher than that in the non-LND group (HR = 1.08 (1.03, 1.13); *p* = 0.001); this result was duplicated in subgroup analysis in advanced-stage epithelial ovarian carcinoma (HR = 1.21 (1.04–1.40); *p* = 0.012). Results were similar in two other meta-analyses [35,36]. Latest meta-analysis showed greater overall survival, but not PFS, in patients with optimally debulked advanced ovarian cancer [37]. Nevertheless, these results must be cautiously interpreted due to the few included randomized trials, which were against systematic LND [36]. However, all of these studies were conducted at the time of primary cytoreductive surgery and for multiple histological subtypes. Initially, the populations of our two groups were heterogeneous, with younger patients in the SL group, with a median age of 59 vs. 67.5 years (*p* < 0.001) and a lower Charlson index (*p* = 0.003) and ASA score (*p* = 0.03). These results agreed with the literature, where patients benefiting from lymphadenectomy are generally younger [10].

The fact that R0 (88.4% vs. 50%; *p* < 0.0001) was highly preponderant in the lymphadenectomy group led to imbalance that could affect survival. This was not in accordance with current recommendations, since almost 10% of patients with SL were not R0 at the end of surgery. R0, even for an expert surgeon according to the study by Eskander et al., had 40% mismatch between surgeon and CT study [38]. In our series, a high CA 125 level at diagnosis was a poor prognostic factor for OS (HR = 1.5, 95% CI 1–2.4, *p* < 0.007; and HR = 1.9, 95% CI 1.1–3.1, *p* = 0.01 in uni- and multivariate analyses, respectively). Pelissier et al. [39] evaluated the ability of CA 125 assay to predict optimal tumor-cytoreduction surgery in patients with advanced ovarian cancer after neoadjuvant chemotherapy. Nearly 70% of patients with advanced ovarian cancer undergo optimal interval debulking surgery after neoadjuvant chemotherapy [40]. In our study, lymphadenectomy was associated with a significant increase in the median duration of surgery similarly to other studies [30,31], but we found no difference in either the number of patients receiving fresh frozen plasma or those transfused there were many missing data for these parameters.

SL did not lead to increased morbidity, since the Dindo–Clavien score was similar between the two groups (*p* = 0.15).

A similar rate of postoperative complications conflicted with LION’s study [10]. Lastly, in this retrospective study, we identified appropriate candidates who benefit from SL. It concerns young, non-comorbid, FIGO Stage III patients with complete response and no more than three cycles of neoadjuvant chemotherapy with R0 surgery at IDS. The main dissemination pathway for ovarian adenocarcinomas is peritoneal [41]. As a result, recurrence of ovarian cancer is essentially peritoneal. A new concept is emerging, hyperthermic intraperitoneal chemotherapy (HIPEC), which would increase the efficiency of surgery by allowing contact with microscopic lesions not seen by the surgeon’s eye. Van Driel et al. [42], in a randomized study in patients with initial Stage III FIGO adenocarcinoma, demonstrated improvement in OS (45.7 vs. 33.9, *p* < 0.05) and PFS (14.2 vs. 10.7, *p* < 0.05), with no difference in morbidity between the two groups. Indeed, it is microscopic tumor residues within the peritoneum after surgery that are responsible for recurrences during patient follow-up [43]. Ceresoli’s study [44] showed that patients with surgery alone had more peritoneal recurrence, 43% vs. 14%. In our study, there was a majority of intraperitoneal and retroperitoneal recurrence in Group 2 compared to Group 1, with no significant difference (*p* = 0.2 and 0.67, respectively). In our study, metastatic implants were found in retrieved lymph nodes despite neoadjuvant chemotherapy including 10% and 13% in pelvic and paraaortic metastasis nodes, respectively. Two studies showed that that the rate of nodal involvement was similar in patients treated before or after chemotherapy, suggesting that nodal metastases are not chemosensitive [20,45]. From a physiological point of view, these results are in favor of systematic LND. The lack of benefits of this procedure, in terms of OS and PFS, highlighted by our study could go against this hypothesis. From a pathophysiological point of view, it could then be interesting to combine both HIPEC and SL to reduce both intraperitoneal and lymph-node recurrences. Attention was also focused on lymphatic-metastasis-associated molecular components associated with focal adhesion, epithelial–mesenchymal transition, and angiogenesis in preclinical models; however, the biology of this rare disease remains unknown [46,47]. The molecular characterization of lymph-node metastasis would be very helpful in clarifying some aspects of this rare pattern of disease in OC natural history, but very few data are available. Earlier flow = cytometric data had fueled the concept that the higher frequency of diploid metastatic lymph nodes in ovarian cancer could result in a lower percentage of S-phase cells, thus sustaining, in principle, worse responsiveness to chemotherapy, and highlighting the role of surgical cytoreduction. Given the balance between survival benefit and surgery-related morbidity during the maximal cytoreductive surgical effort, it is essential to establish optimal selection criteria for identifying appropriate candidates who benefit from surgery without working quality of life, as suggested by the LION trial regarding lymphadenectomy in upfront treatment of advanced OC [10]. This multicentric cohort has good external validation thanks to the population-selection method and multivariate analysis limiting confounding bias. We included patients with advanced ovarian cancer regardless of their initial node status, thus avoiding selection bias. Node removals were routinely performed with a sufficient number of pelvic and lomboaortic lymph nodes (median of 24 nodes) as recommended, thus avoiding performance bias. This study also has some limitations. It is, in fact, a retrospective study, responsible for a significant amount of missing data. The heterogeneity of the practices of each center could explain some of the differences between the two groups. Indeed, the number of patients in the lymphadenectomy group was higher than that in the nonlymphadenectomy group. Lymphadenectomy is a procedure with a considerable treatment burden, and the surgeon’s decision whether to perform such a procedure may depend not only on disease characteristics, such as stage or histology, but also on patient age, performance status, or coexistence [10]. There was indication bias. In fact, the decision to carry out SL depends on the surgeon, who is more likely to perform an additional surgical procedure in young patients without comorbidities than older patients. In the same way, in patients who had undergone SL, there was a very large majority of R0. The impact of SL may be overestimated by the initial characteristics of the patients, and the high proportion of R0 in the SL group, the complete response to NAC, and the number of cycles less than or equal to 3. To avoid heterogeneous surgical quality as a potential weakness in our trial, we introduced the center’s effect in multivariate analyses in OS and PFS, and no significant effects were found.

## 5. Conclusions

In conclusion, we observed that, in a large series of advanced ovarian cancer, in patients receiving SL, the overall survival was not improved after neoadjuvant chemotherapy. On the other hand, SL improved PFS whatever the number of removed lymph nodes and whatever their status (metastatic or not).

The possibility of indication bias cannot be fully discarded given the nonrandomized nature of this study. Indeed, the young age of the patients, the majority of R0, and the complete response to NAC with a number of cycles less than or equal to three may have overestimated the benefit of SL in our study. As a result, the impact of SL might have been underestimated in this analysis. Our findings need to be confirmed by prospective randomized controlled trials with patient characteristics highlighted in this study from which SL could be beneficial.

## Figures and Tables

**Figure 1 jcm-09-02427-f001:**
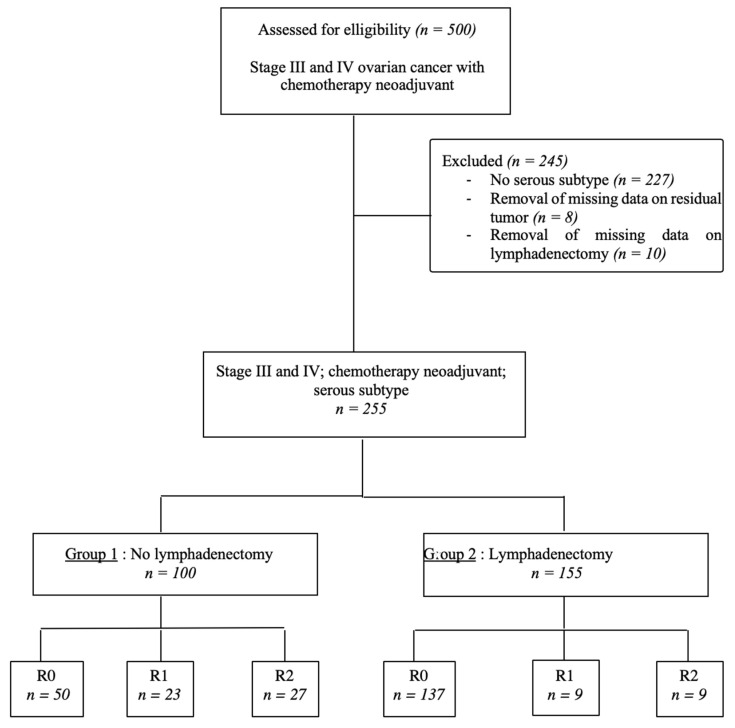
Flowchart of patient selection.

**Figure 2 jcm-09-02427-f002:**
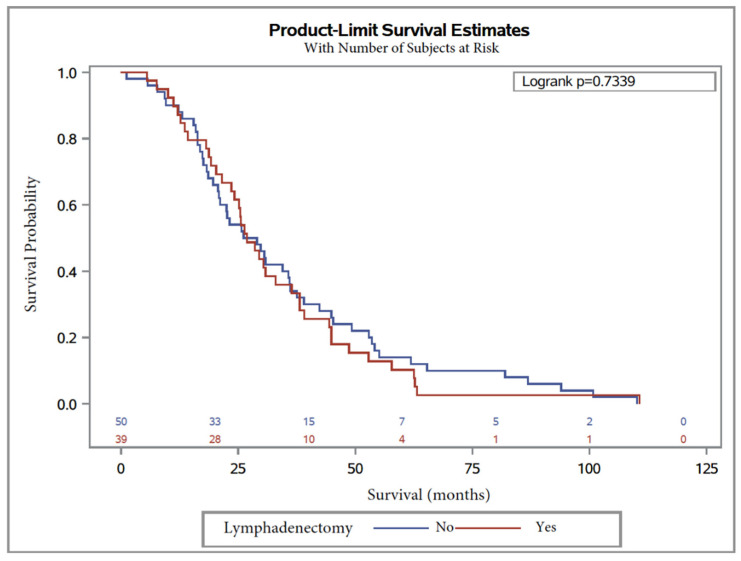
Overall survival in patients with serous ovarian carcinoma after NAC with or without lymphadenectomy.

**Figure 3 jcm-09-02427-f003:**
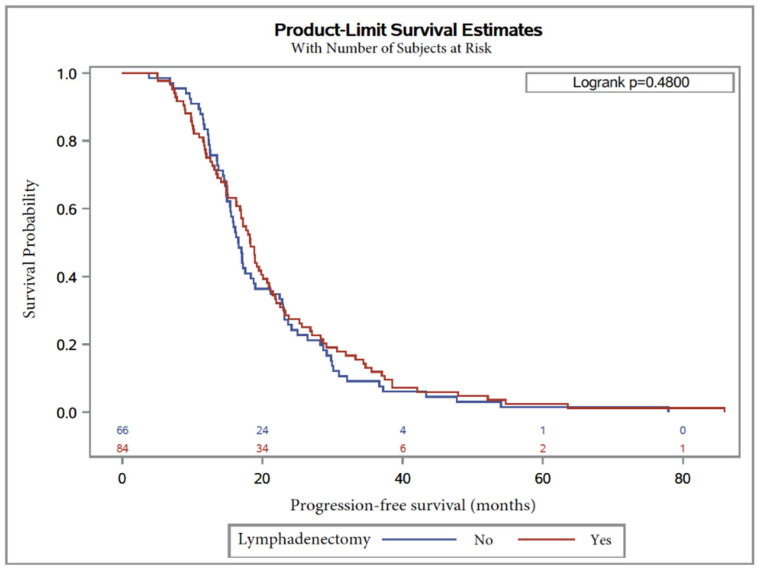
Progression-free survival in patients with serous ovarian carcinoma after NAC with or without lymphadenectomy.

**Table 1 jcm-09-02427-t001:** Patient characteristics.

	Group 1: Nonlymphadenectomy	Group 2: Lymphadenectomy	*p*-Value
Number of patients	100	155	
Age at diagnosis (years)			<0.0001
≤4/9	15 (20.3%)	59 (79.7%)	<0001
50–69	43 (37.7%)	71 (62.3%)	0.009
≥70	42 (62.7%)	25 (37.3%)	0.04
Median (range)	67.5 (31–83)	59 (31–82)	
BMI ^a^ (kg/m^2^)			0.63
<25	38 (49.4%)	70 (54.3%)	
25–30	27 (35%)	37 (28.7%)	
>30 Missing	12 (15.6%) 23	22 (17.1%) 26	
Geographic origin			0.09
Caucasian	74 (93.7%)	74 (89.2%)	
North Africa	1 (1.3%)	7 (8.4%)	
Afro-Caribbean	2 (2.5%)	2 (2.4%)	
Asia	2 (2.5%)	0	
South America	0	0	
Missing	21	72	
ASA ^b^ score			0.03
0	0	2 (100%)	/
1	15 (24.6%)	46 (75.4%)	<0.0001
2	39 (45.9%)	46 (54.1%)	0.45
3	21 (47.7%)	23 (52.3%)	0.76
4	0	1 (100%)	/
Missing	25	37	
Charlson index			0.003
0	11 (18.6%)	48 (81.4%)	<0.0001
≥1	25 (44.6%)	31 (55.4%)	0.4
Missing	64	76	
Gynecological-cancer history			0.9
Personal	3 (3%)	5 (3%)	
None	97 (97%)	150 (97%)	
Genetic mutations			0.19
No mutation	2 (25%)	7 (21.2%)	
Known mutation	2 (25%)	19 (57.6%)	
No research	4 (50%)	7 (21.2%)	
Missing	92	122	

^a^ BMI, body-mass index; ^b^ ASA, American Society of Anesthesiologists.

**Table 2 jcm-09-02427-t002:** Tumoral characteristics.

	Group 1: Nonlymphadenectomy	Group 2: Lymphadenectomy	*p*-Value
**Number of patients**	100	155	
**FIGO Stage**			0.4
III	78 (78%)	127 (81.9%)	0.0008
IV	22 (22%)	28 (18.1%)	0.47
Grade			0.8
1–2	10 (17.9%)	23 (19.3%)	
3	46 (82.1%)	96 (80.7%)	
Missing	44	36	
Preoperative CA125 (U/mL)			0.36
≤1500	58 (65.2%)	102 (70.8%)	
>1500	31 (34.8%)	42 (29.2%)	
Missing	11	11	
NAC ^a^			
Number of cycles			0.02
≤3	5 (5.1%)	25 (16.7%)	
4–6	74 (76.3%)	107 (71.3%)	
≥7	18 (18.6%)	18 (12%)	
Missing	3	5	
Response to NAC			0.08
Complete	5 (27.8%)	19 (52.8%)	0.007
Partial	13 (72.2%)	17 (47.2%)	0.58
Missing	82	119	

^a^ NAC, neoadjuvant chemotherapy; FIGO, International Federation of Gynecology and Obstetrics.

**Table 3 jcm-09-02427-t003:** Operative findings.

	Group 1: Nonlymphadenectomy	Group 2: Lymphadenectomy	*p*-Value
Number of patients	100	155	
Interval bebulking surgery			0.01
Laparoscopy	15 (16.3%)	51 (33.3%)	<0.0001
Laparoscopy with laparoconversion	6 (6.5%)	6 (4%)	1
Laparotomy	71 (77.2%)	96 (62.7%)	0.06
Missing	8	2	
Operative time (min)	242 (155–405)	383 (170–660)	<0.0001
Number of transfused patients			0.7
Yes	9 (60%)	16 (55.2%)	
No	6 (40%)	13 (44.8%)	
Missing	85	126	
Number of transfused red-blood-cell units	1.9 (0–6)	2.2 (0–4)	0.45
Postoperative residual disease (R)			<0.0001
0	50 (50%)	137 (88.4%)	<0.0001
1	23 (23%)	9 (5.8%)	0.02
2	27 (27%)	9 (5.8%)	0.004
Intraoperative complications			0.003
Yes	6 (6.5%)	28 (20.9%)	
No	86 (93.5%)	106 (79.1%)	
Missing	8	21	
Postoperative complications			0.01
Yes	12 (12.8%)	39 (26.7%)	
No	82 (87.2%)	107 (73.3%)	
Missing	6	9	
Dindo–Clavien score			0.15
0	81 (85.3%)	107 (73.8%)	
1–2	10 (10.5%)	29 (20%)	
3–4	4 (4.2%)	9 (6.2%)	
5	0	0	
Missing	5	10	
Number of cycles of adjuvant chemotherapy	6.5 (1–15)	6.9 (2–31)	0.2
Missing	71	106	

Figures given as n (%) or median (range).

**Table 4 jcm-09-02427-t004:** Cox regression analysis for overall survival (OS).

KERRYPNX		UnivariateAnalysis			MultivariateAnalysis *	
Variables	Hazard Ratio	95% CI	*p*-Value	Hazard Ratio	95% CI	*p*-Value
Center			0.4			
1 (ref)	1					
2	1.6	0–123.3				
3	1	0–60.6				
4	0.1	0–1616.8				
5	1.5	0–107.8				
6	0	0				
7	1.8	0–115.6				
8	0.7	0–43				
9	0	0				
Lymphadenectomy: Number of removed lymph nodes						
0 (ref)	1			1		
1–15	0.8	0.4–1.6	0.4	0.7	0.4–1.6	0.4
16–30	1.2	0.6–2.3	0.6	1.1	0.5–2.2	0.7
31–50	1.6	0.8–3.3	0.1	1.8	0.7–4.3	0.06
Age (years) ≤49 (ref) 50–69 ≥ 70						
	1					
	0.8	0.4–1.7	0.7			
	0.8	0.4–1.8	0.7			
BMI ^a^ (kg/m^2^) <25 (ref) 25–30 >30						
	1					
	0.9	0.5–1.5	0.3			
	1.5	0.7–3.3	0.2			
Charlson index			0.4			
0 (ref)	1					
≥1	1.3	0.7–2.5				
ASA						
0 (ref)	1					
1	0.1	0–22.4	0.4			
2	0.1	0–39.1	0.8			
3	0.1	0–26.9	0.5			
4	0	0	0			
FIGO stage						
III (ref)	1					
IV	1.2	0.8–2	0.4			
Grade			0.8			
1–2 (ref)	1					
3	0.9	0.4–1.9				
Preoperative CA125 (U/mL)			0.07			0.01
≤1500	1			1		
>1500	1.5	1–2.4		1.9	1.1–3.1	
Number of cycles						
≤3	1					
4–6	0.9	0.4–2	0.3			
≥7	1.3	0.5–3.4	0.3			
NAC ^b^ response						
Complete (ref)	1					
Partial	1.5	0.5–4.5	0.4			
Surgery debulking						
Laparoscopy (ref)	1					
Laparoscopy with laparoconversion	1.7	0.2–12.6	*0.5*			
Laparotomy	0.6	0.4–1.1	*0.2*			
Postoperative residual disease						
R0 (ref)	1			1		
R1	1.6	0.9–2.8	0.09	1.8	0.9–3.4	0.07
R2	1	0.6–1.6	0.3	1	0.6–1.9	0.4
Number of invaded lymph nodes						
0–5 (ref)	1			1		
6–10	1.2	0.1–21.1	0.4	1.3	0.1–25	0.5
11–20	6.4	0.2–199	0.1	9.6	0.3–320.7	0.06
21–30	1.1	0–33.8	0.6	0.9	0–28.9	0.4
≥31	0	/	/	0	/	/

^a^ BMI, body-mass index; ^b^ NAC, neoadjuvant chemotherapy. * Only variables with *p* < 0.20 in univariate analysis introduced in multivariate Cox regression model.

**Table 5 jcm-09-02427-t005:** Cox regression analysis for progression-free survival (PFS).

		Univariate Analysis			Multivariate Analysis *	
Variables	Hazard Ratio	95% CI	*p*-Value	Hazard Ratio	95% CI	*p*-Value
Center			*0.8*			
1 (ref)	1					
2	1.6	0.1–36.9				
3	0.6	0–11.2				
4	0.2	0–2282.7				
5	0.7	0–16.3				
6	0.5	0–14.1				
7	0.7	0–13.3				
8	0.7	0–13.5				
9	0	0				
Lymphadenectomy: Number of removed lymph nodes						
0 (ref)	1			1		
1–15	1.2	0.7–2	*0.2*	0.2	0.06–0.7	*0.01*
16–30	0.9	0.6–1.4	*0.7*	0.2	0.05–0.8	*0.03*
31–50	0.6	0.4–1.1	*0.1*	0.03	0.04–0.2	*0.0005*
Age (years)						
≤49 (ref)	1					
50–69	0.8	0.5–1.4	0.5			
≥70	0.9	0.5–1.6	0.9			
BMI ^a^ (kg/m^2^)						
<25 (ref)	1					
25–30	1.2	0.8–1.8	0.4			
>30	1	0.6–1.7	0.7			
Charlson index			0.3			
0 (ref)	1					
≥1	0.8	0.5–1.2				
ASA						
0 (ref)	1					
1	3.4	0–951.4	*0.7*			
2	2.9	0–801.9	*0.9*			
3	3.9	0–1103.6	*0.7*			
4	0	0	*0*			
FIGO stage						
III (ref)	1					
IV	1.1	0.8–1.7	0.5			
Grade			0.7			
1–2 (ref)	1					
3	1.1	0.7–1.8				
Preoperative CA125 (U/mL)			0.1			0.1
≤1500	1			1		
>1500	1.3	0.9–1.8		2.8	0.8–9.9	
Number of cycles						
≤3	1					
4–6	1.2	0.7–2.1	0.4			
≥7	1.1	0.6–2	0.9			
NAC response			0.03			0.7
Complete (ref)	1					2
Partial	2.3	1.1–5		1.3	0.3–4.7	
Surgery debulking						
Laparoscopy (ref)	1			1		
Laparoscopy with laparoconversion	2.4	0.9–6.4	*0.07*	154.8	5.4–441.1	*0.003*
Laparotomy	1.1	0.7–1.6	*0.2*	1.4	0.4–4.6	*0.5*
Postoperative residual disease						
R0 (ref)	1			1		
R1	1.7	1.05–2.8	0.1	0.8	0.2–4.4	*0.8*
R2	1.3	0.8–2.1	0.9	1.3	0.1–10.1	*0.8*
Number of invaded lymph nodes						
0–5 (ref)	1					
6–10	0.9	0.5–1.7	0.4			
11–20	1.5	0.5–4.6	0.7			
21–30	1.9	0.4–7.6	0.5			
≥31	1.0	0.1–7.4	0.8			

^a^ BMI, body-mass index. * Only variables with *p* < 0.20 in univariate analysis introduced in multivariate Cox regression model.

**Table 6 jcm-09-02427-t006:** Recurrence characteristics.

	Group 1:Nonlymphadenectomy	Group 2:Lymphadenectomy	*p*-Value
Recurrence			
Pelvic	0	0	
Intraperitoneal	25 (44%)	36 (61%)	0.2
Retroperitoneal	10 (15%)	13 (22%)	0.67
Remote metastasis	23 (41%)	20 (34%)	0.76
Missing	42	86

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
