# Peer review of "Impact of Lymphadenectomy on Survival of Patients with Serous Advanced Ovarian Cancer After Neoadjuvant Chemotherapy: A French National Multicenter Study (FRANCOGYN)"

_jcm, 2020, doi:10.3390/jcm9082427_

Round 1

Reviewer 1 Report

This large, multi-institution study sought to investigate the impact of lymphadenectomy (SL) on survival after neoadjuvant treatment of advanced stage serous epithelial ovarian cancer. This topic is very important clinically and it is not yet fully understood if SL portends a direct survival benefit. The results of such a study could significantly contribute to the management decisions of practicing gynecologic oncologists. Thus, I commend the authors on their attempt to shed further light on a challenging clinical scenario. The methodology employed in this study was also well thought out and appropriate. 

However, I have a few major concerns about this manuscript that need to be addressed before proceeding with publication.

  • The English language used in this manuscript needs major edits. Unfortunately, the diction, grammar and sentence structure used throughout leads to confusion for the reader which may result in misinterpretation of the methodology, results and major conclusions. There is a preponderance of passive voice that should be eliminated and many run on sentences that should be edited for clarity.
    • For example, the abstract contains the following sentences: "(1) Background: This study compared two groups of patients presenting advanced ovarian carcinoma in serous histological subtype benefiting neoadjuvant chemotherapy (NAC) followed by interval debulking surgery (IDS) associated by systematic lymphadenectomy (SL) (group 2) or not (group 1), regarding overall survival (OS) and progression-free survival (PFS) and complications related to surgery... (4) Conclusions: Patients beneficing of SL improved PFS whatever the number of lymph nodes removed and whatever their status (metastatic or not). " This should be rewritten to read "Background: The population of interest to this study was comprised of individuals with advanced stage ovarian carcinoma who were exposed to neoadjuvant chemotherapy (NAC) followed by interval debulking surgery (IDS). Those who did not receive systematic lymphadenectomy (SL) (group 1) were compared to those who received SL (group 2). Outcome measures included progression free survival (PFS), overall survival (OS) and surgical complications... Conclusions: Patients who received SL (group 2) had a significantly higher PFS regardless of node positivity status when compared to those who did not receive SL (group 1)."
    • Lines 73-79 contain sentences that also use confusing language. These sentences need to be edited for clarity.
    • The terms "evidences" and "benefiting" are used improperly throughout the manuscript. Examples can be found in lines 82, 84, 177, etc. "Benefited" should probably be replaced with "received".
    • Line 113: "paraclinical items" should be replaced with "demographic variables".
    • Line 182 needs to be edited for clarity: "It reports operative findings of IDS."???
    • Line 189 needs to be edited for clarity: "while group 1 more often presented a laparotomy (75.6% vs. 59.6% p<0.06)". The term "presented a" is confusing.
    • Line 247 need to be edited for clarity.
    • The above line items are not representative of all of the English language edits that are needed in this manuscript. It is highly recommended that a native English speaker should review and extensively edit the entire manuscript before resubmission.
  • Throughout the manuscript there are too many paragraphs. Many "paragraphs" only contain one or two sentences. This needs to be edited.
  • Did all IDS patients achieve an R=0cm cytoreduction? Line 125 states "All patients had no residual tumor at the end of surgery." However, group 1 has 23 R=1cm and 27 R=2cm subjects. Similarly, group 2 has 9 R=1cm and 9 R=2cm subjects included. 
  • Table 1 includes p values for differences in demographic variables between groups. However, a post-hoc analysis should be performed to identify where the differences are located for those broad categories with significant differences (p<0.05). Consultation with a statistician is strongly suggested.
  • I am very concerned that the PFS results are overstated in the conclusion section and possibly incorrect. Lines 211-212 state: "Similarly, the median PFS was 17.2 months, with 18.3 months (95% CI, 16.3 to 20.1), in group 2 and 16.6 months (95% CI, 14.9 to 18.7) (table 5), in group 1 without any difference between both group logrank 1.1 (0.8-1.5 p=0.48)." Figure 3 also shows no difference in PFS between groups 1 and 2. Therefore, I am unsure where the difference in PFS that is touted in the conclusion section exactly comes from. This needs to be explained before publication can proceed.
  • The discussion restates the results too often and in too much detail (Lines 290-321). The discussion section should be extensively re-written and streamlined. I would encourage the group to re-write the discussion with the question "What do our results mean?" in mind. Instead, the authors have simply restated their results/findings.
  • The conclusions of the manuscript are too strongly stated and should be rewritten to emphasize the differences in R=0cm status and response to NAC status. These difference are likely to be the major factors that explains the difference in PFS. I doubt that SL alone is responsible for the PFS benefit described in this study.

Author Response

The work was corrected by a native English speaker. We also reduced the number of paragraphs.

Not all patients were R0 at the end of the procedure, so we rewrote this paragraph to be clearer. We realized post hoc analysis for age, ASA score, and Charslon's index, which were statistically significant, in Table 1.

Differences in PFS were not present with Kaplan–Meier; it became significant after inclusion in the multiparity model of the variables that had significance threshold < 0.2. This difference is explained by multivariate analysis of the Cox model.

The discussion was rewritten and reduced by removing the repetition of results. Paragraphs that were crossed out correspond to the deleted parts, and highlighted parts correspond to added and/or rewritten parts.

We modified the conclusion by mentioning potentially confounding factors in the study that could also explain improved survival, namely, the majority of R0, the reduced number of chemotherapy courses, young age, and complete response to neoadjuvant chemotherapy.

Reviewer 2 Report

General comments

This paper describes an observational study of 255 women with advanced (stage III-IV) serous ovarian cancer who received neoadjuvant chemotherapy followed by interval debulking surgery, comparing outcomes between those who underwent lymphadenectomy and those who did not. The authors found that those treated with lymphadenectomy had improved progression-free survival compared to those who did not undergo lymphadenectomy, but there was no significant difference in overall survival.

In general, I thought this was a good paper: the methods of the study were sound and clearly-described; the analysis was appropriate, and the interpretation was reasonable, with some of the key limitations highlighted.

I have made comments below on possible areas for development or modification, most of which are fairly minor. I have also suggested some changes to English phrasing for style and clarity.

Scientific comments

Potential for bias and confounding: The authors acknowledge in their discussion that the observational nature of this study means there is a considerable potential for bias and confounding – but I think this issue might need even more explicit reflection. In particular, as they note in the results, patients who did not undergo lymphadenectomy tended to be older and have more comorbidities compared to patients who did undergo lymphadenectomy. This is likely to be because surgeons are more cautious about doing major surgery on patients who are older and more frail (as the authors say towards the end of the discussion section). It is important to account for these differences as much as possible, as they could well lead to differences in outcome, quite apart from lymphadenectomy.

I understand that the authors only included variables in the multivariable regression model if they reached a certain significance threshold, on which basis they excluded several variables, including age. However, I think many people would consider variables such as age and stage at diagnosis as very strong a-priori confounders. It might be reassuring to also give results of, perhaps, a sensitivity analysis in which all these other factors are also adjusted for?

In the conclusion, the authors suggest that due to “the possibility of an indication bias”, as outlined above, “the impact of SL might have been underestimated in this analysis”. But is the bias not likely to be the other way round? If younger, fitter people are more likely to have a lymphadenectomy (as has been shown here), then surely we would expect them to have better survival (perhaps such as the improved PFS seen here) even if lymphadenectomy had no survival benefit?

English language/ style

The manuscript was in general very clearly written, but I would suggest a few modifications. In particular, I think the authors’ use of the phrase ‘benefiting from’, when they mean ‘treated with’ or ‘underwent’, is potentially confusing. [In the context of treatments, ‘benefit from’ tends to imply ‘had a better outcome as a result of’ – which is generally not what the authors mean when they use the phrase here]. This appears in multiple locations throughout the manuscript (e.g. lines: 35, 46, 84, 98, 105, 168, 177, 189, 293).

In addition, I have listed below some other phrases I would suggest changing for style or clarity [with altered words in square brackets].

  • Lines 34-36: “This study compared two groups of patients presenting [with] advanced ovarian carcinoma [of] serous histological subtype [who were treated with] neoadjuvant chemotherapy (NAC) followed by interval debulking surgery (IDS), [with additional] systematic lymphadenectomy (SL)….”
  • Lines 41-42: “Patients were [on average] younger and less comorbid…”
  • Line 59: “R0 is recognized as a major prognostic factor [for] increased overall survival…”
  • Line 71: “In contrast, two [prospective] randomized trial[s] did not show an overall survival benefit [10,19], [though one found] an increased progression free survival [19].”
  • Line 73: I think this should be “extent” not “extend”?
  • Line 82: “…the absence of a consensus [regarding] systematic lymphadenectomy…”
  • Line 84: “…patients presenting [with] primary serous ovarian carcinoma…”
  • Line 127: “The residual tumor was [assessed] at the end of IDS.”
  • Lines 163-164: “…155 patients in the [lymphadenectomy] group (60.8%) and 100 patients in the [no-lymphadenectomy] group (39.2%).”
  • Line 173: “In the [lymphadenectomy group] (table 2), the proportion of patients…”
  • Line 182: “It reports operative findings of IDS”. I’m not quite sure what this means? I don’t think the phrase is clear as to what ‘it’ refers to?
  • Lines 195-196: “…the Dindo-Clavien score was [not] statistically significant…”.
  • Line 203: “…with a number of cure similarly (median 6.5 vs 6.9 p=0.2).” I’m not quite sure what this means? I think it may be referring to Table 3, which I think shows a similar average number of cycles of chemotherapy – but I don’t see any numbers given of patients ‘cured’ (‘cure’ would be rare in advanced ovarian cancer anyway).
  • Line 208: “The logrank test [showed no significant] difference…”
  • Line 209: “In univariate [Cox regression analysis], lymphadenectomy was not statistically [significantly associated with] overall survival with HR=0.9 (0.6-1.4), p=0.7.
  • Lines 215-216: “In multivariate analysis, we found that [CA125 level at diagnosis was a predictor for worse OS] [HR=1.9 (1.1-3.1), p=0.01] (table 4).”
  • Line 237: “In our study, we found that patient[s] with initially inoperable…”
  • Lines 269-270: “Other studies [have] shown that systematic lymphadenectomy at the time of primary cytoreduction [was not associated with improved OS or PFS [10, 33]].”
  • Lines 283-284: “[Results were similar in two other meta-analyses [35,36]].”
  • Line 301: “In our series, high [CA 125 level at diagnosis] was a poor prognostic factor for OS.”
  • Line 309: “In our study, [lymphadenectomy was associated with a significant increase in] the median duration of surgery…”
  • Line 310: “…but we [found] no difference [in either] the [number of] patients [receiving fresh] frozen plasma…”
  • Line 315: “The rate of laparoconversion is [not] statistically significant…”
  • Line: 318: “Clavien score was [not] statistically significant…”
  • Lines 333-334: These lines refer to the study by Delangle et al, and the finding that the number of involved pelvic lymph nodes was an independent predictor of isolated lymph node recurrence (p.67 in the Delangle paper), but the phrasing and relevance seem a little unclear to me?
  • Lines 352-353: “…the number of patients in the lymphadenectomy group was higher [than] in the non-lymphadenectomy group.”
  • Line 366-367: “Our findings need to be confirmed by prospective [randomized] controlled trials.”
  • Table 1: I think “Age at diagnostics” should perhaps be “Age at diagnosis”?
  • Table 3, final section: This is labelled “Number of adjuvant chemotherapy” – I think it might be clearer to label it “Number of cycles of adjuvant chemotherapy”.
  • Table 4: I think that in a couple of sections of this table, the lines of numbers have become misaligned, and moved one line down (see “Number of lymph nodes removed”, “Preoperative CA125”, Postoperative residual disease, and “Number of lymph nodes invaded”).

Author Response

The work was corrected by a native English speaker. We insisted on the discussion on the choice of patients at time of surgery of whether to perform lymphadenectomy.

We chose to set a threshold for the entry of variables into the multivariate model. After reading your comments, we added age and FIGO stage to diagnosis in our model. We obtained similar results with improvement of the PFS considering age and FIGO stage. In this context, we left the model as it was initially, since we showed that these two criteria are not confounding factors, and that there is a real relationship between lymphadenectomy and improved PFS.

In the conclusions, we reworded the sentence, because patients with a majority R0 level in the cure group may have been overestimated, and the fact that patients showed better response to chemotherapy with a lower number of cures.

With regard to the relevance of line 354–357, we withdrew the paragraph in question since we are already discussing recurrence sites.

Reviewer 3 Report

The role of lymphadenectomy in advanced ovarian cancer has always been highly debated both in upfront surgery, in interval surgery and in recent years also in recurrence.

Attention has also been focused on the lymphatic metastasis-associated molecular components associated with focal adhesion, epithelial-mesenchymal transition, and angiogenesis in preclinicalmodels; however, the biology of this rare disease remains stillunknown.The molecular characterization of lymph node metastasis would be very helpful inclarifying some aspects of this rare pattern of disease in OC natural history, but very few data are available. Earlierflow cytometric datahad fueled the concept that the higher frequency of diploid metastatic lymph nodes in ovarian cancer could result in a lower percentage ofcells in S-phase, thus sustaining, in principle, worse responsiveness to chemotherapy, and highlighting the role of surgical cytoreduction.

Given balance between survival benefit and surgery-relatedmorbidity during the maximum cytoreductive surgical effort, it is essential to establish the optimal selection criteria for identifying appropriate candidates who will benefit from surgery without working quality of life, as suggested by the LION trialregarding lymphadenectomy in upfront treatment of advanceD OC

In this sense it seems very interesting to report in the discussion some studies that explain pathological and clinical anatomical corrections with some very specific lymph node diffusion pathways.

It would be useful to mention some works in this regard:

Hepatoceliac Lymph Node Involvement in Advanced Ovarian Cancer Patients: Prognostic Role and Clinical Considerations. Ann Surg Oncol. 2017;24(11):3413-3421. doi:10.1245/s10434-017-6005-1

Mesenteric lymph node involvement in advanced ovarian cancer patients undergoing rectosigmoid resection: prognostic role and clinical considerations. Ann Surg Oncol. 2014;21(7):2369-2375.

Author Response

We thank you for these pathophysiological explanations that allowed us to improve our discussion. We also integrated the two articles in the bibliographic references.

Round 2

Reviewer 1 Report

Drastically improved presentation of a very interesting study. Well done.